# In Vitro and Human Pilot Studies of Different Topical Formulations Containing *Rosa* Species for the Treatment of Psoriasis

**DOI:** 10.3390/molecules27175499

**Published:** 2022-08-26

**Authors:** Diana Ioana Gavra, Laura Endres, Ágota Pető, Liza Józsa, Pálma Fehér, Zoltán Ujhelyi, Annamária Pallag, Eleonora Marian, Laura Gratiela Vicas, Timea Claudia Ghitea, Mariana Muresan, Ildikó Bácskay, Tünde Jurca

**Affiliations:** 1Department of Pharmacy, Faculty of Medicine and Pharmacy, University of Oradea, 1st December Square 10, H-410028 Oradea, Romania; 2Doctoral School of Biomedical Sciences, University of Oradea, H-410073 Oradea, Romania; 3Department of Psycho-Neuroscience and Recovery, Faculty of Medicine and Pharmacy, University of Oradea, 1st December Square 10, H-410073 Oradea, Romania; 4Department of Pharmaceutical Technology, Faculty of Pharmacy, University of Debrecen, Nagyerdei Körút 98, H-4032 Debrecen, Hungary; 5Doctoral School of Pharmaceutical Sciences, University of Debrecen, Nagyerdei Körút 98, H-4032 Debrecen, Hungary; 6Institute of Healthcare Industry, University of Debrecen, Nagyerdei Körút 98, H-4032 Debrecen, Hungary; 7Department of Preclinical Discipline, Faculty of Medicine and Pharmacy, University of Oradea, 1st December Square 10, H-410073 Oradea, Romania

**Keywords:** phytochemical profile, antioxidant capacity, bioactive compounds, topical formulation, human clinical study, biocompatibility

## Abstract

The aim of this study was to evaluate the phytochemical profile and antioxidant properties of the extracts from three *Rosa* species (*R. canina*, *R. damascena*, *R. cairo*), to develop and investigate topical formulations with lyophilized forms of extracts for the treatment of psoriasis. Phytochemical screening and in vitro total antioxidant capacity (DPPH, FRAP, CUPRAC, SOD) of studied samples were examined and compared. Lyophilized extracts of roses were dissolved in Transcutol HP and different formulations of creams were prepared. Franz diffusion method was used to evaluate the drug release and biocompatibility was tested on HaCaT cells. *Rosa damascene* had the best results regarding all the analyses that were conducted. After the evaluation of topical products, the formulation with *Rosa damascena* extract in a self-emulsifying drug delivery system was tested on a human clinical study that involved 20 patients. At the end of the clinical study an improvement in the quality of life of the patients was observed and erythema, induration and scaling were reduced. The present study indicates that our examined extracts exhibited great phenolic content, antioxidant capacity and safety profile of topical formulation and therefore can be used as a reliable source of natural antioxidants and may be used as a complementary treatment to improve the quality life of patients with psoriasis or may be tested on another diseases.

## 1. Introduction

Nature has been a source of medicinal products for millennia, with many useful drugs developed from plant sources [1]. Rose is one of the most important groups of ornamental plants and their fruits and flowers are used in a wide variety of foods, nutritional products and different traditional medicines. Various components were isolated from flowers, petals and hips (seed-pot) of *Rosa* sp. including terpenes, glycosides, flavonoids, and anthocyanins [2]. The *Rosaceae* family is an old family of roses and contains over 2500 species in more than 90 genera. *Rosa damascena* is the vital species of the *Rosaceae* family and it is especially famous for its perfuming properties and pharmacological activity [3,4]. Many compounds are present in the *R. damascena* as flavonoids, anthocyanins, terpenes and glycosides, and polyphenolic acids, which have therapeutic effects [4]. Analyses performed over time by various researchers have shown that *R. damascena* has analgesic [5], anti-inflammatory, antioxidant [6], antibacterial, antiviral, antimicrobial [7], anticonvulsant [8,9], antidepressant, anticancer, relaxing and hypnotic effects. It is also used for various digestive problems and has been shown to be useful in eye diseases [10]. Products containing plant extracts are increasingly used in the treatment of dermatological conditions such as psoriasis. Psoriasis is a chronic inflammatory skin disease characterized by erythrosquamous plaques, skin dryness, and various degrees of pruritus [11]. Epidermal hyperplasia, inflammation can be inhibited by their compounds, so they can be used to treat this condition. However, information on the efficacy and safety of these topical products is sometimes insufficient in most studies [12]. As a result, more scientific evidence is needed to promote the topical treatment of psoriasis using plant extracts, which must be justified by reliable clinical trials.

With its intermittent episodes of exacerbations and remissions, psoriasis has a substantial impact on physical health and quality of life of the affected individuals, which requires lifelong management [13]. Conventional treatments include topical treatment, phototherapy, and systemic treatment, which have different degrees of therapeutic efficacy and adverse reactions. Despite the good response to biological agents, relapse is inevitable after withdrawal of therapy. Improvement of epidermal function with topical emollients can mitigate and/or prevent the relapse of psoriasis [14]. Therefore, long-term skin care could be an effective and economical way to prolong the therapeutic effects of topical or systemic treatments and delaying relapse. Conventional therapies have a limited efficacy and may cause a number of side effects, which may include cutaneous atrophy, organ toxicity, carcinogenicity, and broadband immunosuppression. Because of the side effects caused by the conventional therapies it would be desirable to use herbal products as an alternative treatment for psoriasis [15,16].

In order to estimate the correlations between chemical compounds and therapeutical effects, three different *Rosa* species were examined in this study. The aim of this study was to evaluate the phytochemical profile of the extracts, to develop and investigate topical formulations with lyophilized forms of three *Rosa* species (*Rosa damascena*, *Rosa canina* and *Rosa cairo*). Before choosing the three types of roses, we made a comparative analysis on 11 different species regarding the total content of polyphenols, flavonoids and antioxidant activity. The three species were harvested from rose growers, but no studies and informations of on *Rosa cairo* are found in the literature, as this is a new species. After evaluating the results, we decided to conduct a clinical study to verify the effectiveness of the cream with *Rosa damascena* extract in patients suffering from psoriasis. We chose this extract because it has the best antioxidant activity, is rich in polyphenols, and we tried to increase the drug solubility and dissolution properties preparing a topical formulation with SNEDDs that showed the best biocompatibility and diffusion that were analyzed in the preformulation stage. In the present work in vitro antioxidant activity was assessed by SOD assay. MTT method was used to verify the cytotoxicity of the samples.

## 2. Results

### 2.1. Determination of the Components and the Antioxidant Capacity of the Extracts

Qualitative characterization of the extracts was performed by HPLC method. Identified compounds for *Rosa damascena* were presented and discussed in a previous article [17].

Using an HPLC assay, 10 compounds were identified in *Rosa canina* extract, as isoquercetin, found in the highest amount (188.053 µg/mL), followed by quercetitrin (129.927 µg/mL) and kaempferol-3-rhamnoside (81.539 µg/mL). Other identified compounds were quercetol, kaempferol, rutoside, chlorogenic acid, 4-o-caffeylquinic acid, caftaric acid and p-coumaric acid in concentrations between 11.185 and 1.1946 µg/mL. On the other hand, only six polyphenolic compounds, namely kaempferol-3-rhamnoside (537.624 µg/mL), quercetrin (370.916 µg/mL), hyperoside (162.320 µg/mL), rutoside (83.543 µg/mL), isoquercetrin (58.911 µg/mL) and kaempferol (19.785 µg/mL) were identified in the extracts obtained from *Rosa cairo*.

Using Folin-Ciocalteu reagent, total phenolics content was determined total phenolics content and was expressed as mg equivalent of gallic acid (GAE) per mL extract, In *Rosa canina* extract was found a total phenolic content of 117.69 mg GAE/100 G dw was found. For *Rosa cairo* the amount obtained by us was 116.95 mg GAE/100 G dw.

The results for total flavonoids content were obtained using a colorimetric method AlCl_3_. The results were expressed as mg equivalents quercetin (QE) per mL extract according to the calibration curve. Were obtained 1.96 mg QE/mL extract for *Rosa canina* and 1.55 mg QE/mL extract for *Rosa cairo*.

Regarding the antioxidant activity, the results presented in Table 1 showed variable but high values and were studied by DPPH, FRAP and CUPRAC methods. The antioxidant activity of ethanolic extracts was mainly due to the high level of total phenolic content. These results provide a theoretical basis for potential explorations of its pharmacological activities.

### 2.2. Investigation of Self-Nano-Emulsifying Drug Delivery System

Pseudoternary phase diagrams were constructed by using a conventional water titration technique. Isopropyl-myristate as oily phase, Cremophor RH 40 as surfactant and Transcutol HP as co-tenside were used in the formulation. The maximum nanoemulsion existing zones observed are presented in Figure 1. According to the diagrams the most favorable formulation is the one, which contained *R. damascena* extract because of the largest nanoemulsion existing zone.

The distribution of droplet size and the zeta potential of each composition were also determined and presented in Table 2. According to the DLS and the zeta potential measurements all of the compositions are in the nano-size range and have negative zeta potential. The developed rose-containing SNEDDS spontaneously formed from nanoemulsion upon mild agitation in distillated water at room temperature. The concentration of the different rose extracts was 50 mg/mL in each composition. The formulated SNEDDS were stable for three months at room temperature.

### 2.3. Ointment Formulation

Ointments were prepared containing the different rose extracts, in suspended (lyophilized) form or in self-emulsifying drug delivery system. Six compositions were formulated. The ointments were manufactured at the laboratory scale considering the final batch quantity and process parameters of the products. The exact compositions are demonstrated in the Materials and Methods section.

### 2.4. Cell Viability Study (MTT Assay)

Results of MTT assay are presented in Figure 2. During the experiments, PBS was chosen as a negative control and TritonX-100 as a positive control. Cell viability values are compared to PBS and expressed as the percentage of the negative control. In case of ointment formulations of *Rosa canina*, *Rosa damascena* and *Rosa cairo*, the compositions are safe and well tolerated by the cells, cell viability is over 70% in each case, which corresponds to ISO recommendation (ISO 10993-5 recommendation). However, in the case of SNEDDS samples cell viability values are decreased (Figure 2). In the case of *Rosa damascena* SNEDDS, cell viability is over 70%, *Rosa cairo* SNEDDS reaches 70%, but *Rosa canina* SNEDDS is under 70% of viable cells. The lower cell viability results may be due to the high surfactant content of the formulation.

### 2.5. Texture Analysis

Compression tests were evaluated with different ointment compositions. The compression forces (N) required to insert the probe to a given distance into formulations are shown in Figure 3. According to our investigation, different compositions need different amounts of force. Formulations containing the rose extract in lyophilized form and in the self-emulsifying system were compared to each other. The results showed that ointments containing the active ingredient in SNEDDS demonstrated lower resistance to deformation compared to those formulations where rose was in suspended form. A lower resistance value indicates a lower level of firmness, which is a desirable factor for spreading on the skin. Higher compression values indicate a hard cream consistency that may hinder the liberation of active agent. According to the results, ointments containing SNEDDS are more favorable compositions.

### 2.6. In Vitro Release Studies

Figure 4 and Figure 5 show the cumulative amount (%) of diffused quercetin in the different formulations as a function of time (minutes). The highest diffused amount of quercetin was obtained in the formulations with SNEDDS. The maximum diffuse amount of quercetin was 31.83% after 90 min, and in the formula without nanoparticles, the maximum amount of diffused quercetin reached 19.54% after 60 min, remaining at the same value after 90 min, in preparations containing *Rosa damascena* extract. In creams with and without nanoparticles, which contained *Rosa cairo* and *Rosa canina* extracts, the diffusion differences of quercetin were relatively small. In Table 3. quercetin release rate and the diffusion coefficient values related to the compositions are presented.

### 2.7. Superoxide Dismutase Activity of the Topical Formulations

In Figure 6, in vitro SOD activity is expressed as a percentage of SOD activity in untreated keratinocytes. UVB irradiated cells pretreated with PBS were used as a negative control. For positive control (±)-6-hydroxy-2,5,7,8-tetramethylchromane-2-carboxylic acid (Trolox) dissolved in PBS was used in pretreatment at a concentration of 15 µM. The SOD enzyme activity of the non-UV group was taken as 100%. The SOD activities of the treated groups were compared with the enzyme activities of the controls, which were pretreated with the compositions with or without SNEDDS. 

According to the results, ointment containing rose extract in SNEDDS formulation showed a significant increase in free radical scavenging activity. Pretreatment with the ointment containing the *R. damascena* in SNEDDS showed the highest SOD enzymatic activity (30%) as we can observe on the statistical significance (*p* < 0.001) compared to the other rose extracts (SOD activity < 30%; *p* < 0.0001).

### 2.8. Clinical Investigation

A statistical description of the research parameters was made, which shows a symmetrical distribution, following the Skewness and Kurtosis coefficient.

Following the surface of the affected area, in the two groups of patients studied, at the first and second visit, nine people were observed with an impairment of <10%, 10 persons with an impairment of 10–29% and one person with a damage of 30–49% of the limb surface. In the third visit, in three patients, a total remission was observed with an impairment of 0%, in three patients there was an impairment of <10%, and in the case of impairments in the category of 10–29%, 30–49%, a decrease was observed, but insignificant, and insufficient to be able to fall into the lower category. Following treatment, we can see an improvement in patients’ quality of life using PASI and DLQI values. The assessment of the effectiveness of the treatment was carefully monitored by the three elements of the PASI score: erythema, induration, desquamation, whose values gradually decrease after the 3 visits performed from the moment of starting the treatment.

The description of the affected area was followed at the three visits, depending on the percentage of the affected area. Thus, four categories were followed, the first with “0%”, the second “>10%”, the third category between “10–29%”, and the fourth “30–49%” of the affected area.

From Figure 7, a decrease in the values of the parameters from visit 1 to visit 3 can be seen, a decrease that is considerable for the first group, as opposed to the placebo group, where there is an insignificant decrease. The area affected in patients who used the extract cream is diminished after the treatment period. The interpolated line also shows a decreasing trend.

In Figure 8 we can follow the three elements of the PASI score over the 6 weeks, erythema, induration and desquamation, and we can see a more significant improvement in research Group 1.

From Table 4, where the comparative statistics of the parameters between the three visits are presented, it can be observed:

-The area does not differ significantly between visit 1—visit 2, nor between visit 1—visit 3, nor between visit 2—visit 3, *p* > 0.05.

-Erythema differs significantly between visit 1—visit 2, visit 1—visit 3, but also between visit 2—visit 3, *p* < 0.05.

-Desquamation also differs significantly between visit 1—visit 2, visit 1—visit 3, but also between visit 2—visit 3, *p* < 0.05.

-Induration also differs significantly between visit 1—visit 2, visit 1—visit 3, but also between visit 2—visit 3, *p* < 0.05.

The relationship between the two parameters (induration and desquamation) is directly proportional, and strong. As induration decreases, so does desquamation. This statistically significant link can be seen both from the value of the Pearson coefficient in Table 5 and from the graphical presentation of Figure 8.

The area correlates positively, directly proportionally, statistically significant with erythema and induration, DLQI which can also be seen in Table 5, using the value *p*. Erythema correlates significantly with area, induration and desquamation, PASI and DLQI which shows the value of *p* < 0.05. We can also observe a positive relationship between desquamation and erythema, induration, and PASI score.

Induration is directly proportional to the area and erythema, as can be seen in the figure below. The positive correlations of PASI are significant with erythema, desquamation and induration. Between DLQI and arena, erythema, and PASI, there is a directly proportional relationship given by *p* < 0.05. As a result of the positive relationship, we can see that as the erythema decreases, so does the induration.

In patients who used *Rosa damascena* cream, at the first evaluation (after two weeks of use) a reduction of desquamation and induration was observed. After 6 weeks of use, all three indices (erythema, induration, scaling) of the assessment tool (PASI score) were reduced. At the end of the 6 weeks of use of the cream, an improvement in the quality of life of the patients could also be observed following the DLQI score.

For patients who used placebo, a slight reduction in scaling was observed at the first assessment. Erythema and induration were less reduced in contrast to patients who used the cream with extract.

## 3. Discussion

Medicinal plants are an important source for therapeutic products which can prevent or relieve numerous illnesses as they are an important source of phytochemicals and other bioactive compounds. Reactive oxygen species (ROS) are implicated in a large number of illnesses, especially chronic ones. Nitrogenous species and free radicals start chain reactions which can favour the initiation and progression of many complications in diseases [18].

The fruits of the rose species are considerably beneficial for human health since they contain organic and inorganic matters. The fruits of the rose species, are rich in minerals, vitamins, sugars, phenolic compounds, carotenoids, tocopherol, bioflavonoids, tannins, organic acids, fruit acids and volatile oils [19]. 

Qualitative characterization of the extracts was performed by HPLC method. Qualitative, quantitative characterization, and antioxidant activity for the alcoholic extract of *Rosa damascena* can be consulted in our previous research article as phytochemical screening and biological activity shows that the extract contains is a wide range of natural sources which can be used for the discovery of natural pharmaceuticals and due to the high content of bioactive compounds (TPC = 321 ± 0.03 mg GAE/100 g DW and TFC = 32.4 ± 0.01 mg QE/100 g DW) and high values for antioxidant activity: DPPH (94.11 ± 0.01%), CUPRAC (555.316 ± 0.06 µmol Trolox/mL), FRAP (354.43 ± 0.03 µmol Trolox/mL) and ABTS (76.351 ± 0.02 µmol Trolox/mL), it can be used as a prophylactic agents or as an adjuvant treatment in the pathologic process of various disorders. [17]. 

Analyses of *Rosa canina* fruits revealed that phloroglucinol, gallic acid, protocatechuic acid, chlorogenic acid, caffeic acid, p-coumaric acid, ferulic acid, catechin, epicatechin, quercetin, quercetin-3-glucoside, apigenin, resveratrol and kaempferol were identified [20]. The HPLC analysis reported in literature showed that the *R. canina* extract was especially rich in the polyphenols as hyperoside, astragalin, rutin, (+)-catechin and (−)-epicatechin as well as in the terpenoid ursolic acid and a poly-hydroxylated organic acid organic acid, which is the quinic acid [21]. Several other polyphenols have been identified in trace amounts in *R. canina* extracts such as ellagic acid, salicylic acid, vanillic acid, ferulic acid and caffeic acid [22]. Our results are in accordance with other researchers. Quercetin is the most abundant flavonoid that accumulates in superior plants where it forms glycosides, such as rutin, with a great variety of sugars [23]. Quercetin and its derivative quercetin-3-O-glucuronide, inhibit ROS overproduction making chemoprotection of mitochondrial function through antioxidative action [24]. Other studies showed that a compound isolated from *Rosa canina* rose hips as quercetin was a potent melanogenesis inhibitor in B16 mouse melanoma cells [25].

In our work, we can observe that analyzed species of *Rosa* can be a natural source of antioxidants with a potential use in disease prevention and treatment.

In *Rosa canina* extract we found higher value compared to other authors that reported total phenolic content in 50 % (*v*/*v*) ethanol extracts from *Rosa canina* L. fruits—69.4 mg GAE/g dw, the lowest value was found in the water extract—55.4 mg GAE/g dw [26]. In previous studies for ethanolic flowers extract was reported a higher number of flavonoids (2.372 mg QE/mL) were reported, using an aluminium chloride colorimetric method and quercetin was used as a calibration standard. Many studies are focused on finding the relationship between a flavonoid’s structure and its antioxidant activity [27,28].

To evaluate antioxidant capacity we used DPPH, FRAP and CUPRAC methods. The results are presented in the table above. The positive correlation between the total phenolic content and the antioxidant activity evaluated by DPPH, FRAP and CUPRAC assays was observed. CUPRAC assays, based only on single electron transfer mechanism were considered as the more relevant method for evaluation of antioxidant potential of extracts, because the reaction conditions are close to the pH of human blood serum. Our results for CUPRAC were higher than the report of Taneva et al., which obtained the value—1358.2 mM TE/g, but instead they obtained higher value for DPPH [26].

There are data showing that methanol/water extract of fresh rose petals or flowers from different countries have a dose-dependent anti-radical effect against 2,2-diphenyl-1-picrylhydrazyl (DPPH) radicals, significantly higher than that of standard vitamin E, BHT, and BHA [29,30,31]. In vivo tested antioxidant properties of different aqueous extracts from rose flowers demonstrated ferric-reducing antioxidant power ability close to that of the control group [29]. Among the fractions obtained by extraction with organic solvents (petroleum ether, chloroform, acetone, methanol, and water) of lyophilized powder from the fresh juice of *R. damascena* Mill. flowers, the acetone fraction inhibits 50% of the superoxide anion radical (•O^2−^) production, hydroxyl radical (•OH) generation, and lipid peroxidation, in a concentration dependent manner [32]. 

Recently, researchers have shown interest in lipid-based formulations because of their ability to deliver a wide variety of drugs to different parts of the body. Nanoemulsion-based delivery routes, especially self-nanoemulsifying drug delivery systems (SNEDDSs), have been reported as a suitable approach for enhancing the bioavailability and bioactivity of various drugs or natural bioactive compounds. SNEDDSs include a uniform mixture of oil, surfactant, co-solvent and drug substance and form a nanoemulsion upon dispersion under agitation, when diluted with aqueous solutions. SNEDDSs provide advantages, such as production of small-sized droplets of oils, resulting in a high surface area, increased gastrointestinal tract membrane permeability, enhanced drug loading efficiency and protection of the drug from degradation in the gastrointestinal tract [33,34,35]. 

There are selection criteria for the components of an SNEDDS as safety (biocompatibility)–the most cytotoxic component of an SNEDDS is the surfactant [36,37]. Compared to ionic ones, nonionic surfactants are less toxic; the value of hydrophilic-lipophilic balance (HLB) of the surfactant and data from literature shows that this value for the formation of a nanoemulsion should be > 10 [38]. HLB for Cremophor RH 40 is 15 favoring the SNEDDS formation [39]. A suitable co-surfactant: this component, accompanied by the surfactant, reduces the interfacial tension of the nanoemulsion. Co-surfactant must be solubilized in oil and surfactant. PEGs are biocompatible species with both hydrophilic and hydrophobic properties, which is why we choose Cremophor RH 40 (PEG-40 castor oil) as a co-surfactant. The composition of SNEDDS includes isopropyl myristate, Cremophor RH 40 and Transcutol HP.

In our preparation we have used sucrose stearate SP 70 as an emulgent. Sucrose esters show many advantages as penetration enhancers via biodegradability and lack of toxicity. An emulsifier system comprising a mixture of sucrose esters is able to form multilamellar liquid crystalline networks, which possess an excellent safety profile and display a broad range of formulation applications [40]. Some researchers developed nanoemulsions with improved structure and long-term stability by employing a natural sucrose stearate as a sole surfactant. In their study, a thorough comparison between the novel sucrose stearate-based nanoemulsions and the corresponding lecithin-based nanoemulsions revealed that the SE was superior in terms of emulsifying efficiency, droplet formation and physical and chemical stability. These surfactants may play a role in solubilization and may also modify the bioavailability of drugs. Animal studies have demonstrated that the SEs display neither toxic nor carcinogenic activity and can be considered safe excipients in topical and oral formulations [41,42]. 

After establishing the components of SNEDDSs, optimization studies are conducted to obtain the best amounts of oily phase, surfactants and cosolvents that generate spontaneous nano-emulsion. Ternary phase diagrams are used for the identification of the emulsification area using the selected components. This area can be identified visually or by measuring the particle size of the nano-emulsion that results after the aqueous dispersion. It was shown that the SNEDDSs’ composition from the emulsification area generate spontaneous nanoemulsions that have globule sizes less than 200 nm. After aqueous dispersion we can observe a region with homogenous and clear systems that is represented by the region of nano-emulsification. According to our diagrams the most favorable formulation is the one which contained *R. damascena* with the largest nanoemulsion existing zone and the smallest droplet size (91.75 ± 4.12 nm).

To prove the safety of the formulations, cytotoxicity evaluations were realized on the HaCaT cell line. MTT assay is one of the most used techniques for preliminary assessments of the antiproliferative activity of bioactive compounds. MTT cytotoxicity assays were performed on HaCaT cells with each sample prepared with Franz diffusion cells, using PBS as the receptor phase. Cell viability tests were performed on each extract, and cell viability for nanoparticle preparations was also assessed. 

The MTT assay has gained much attention due to the effect of its numerous advantages. Properly executed, it allows us to obtain the results with an accuracy similar to the one of traditional plate counts with supreme rapidity, but a variety of chemical compounds are known to interfere with the MTT assay, and these are mostly reducing compounds that lead to non-enzymatic reduction of the MTT to formazan. The MTT assay classifies as one of the cheapest and simple tests—considering reagents and equipment required. Another noticeable advantage is the applicability of the microplate assay, useful both in advanced experiments and screening assays [43]. This method is versatile and overcomes the solubility issues associated with compounds and extracts. It could be developed as a high-throughput assay for determining the antioxidant potential of large volumes of herbal samples. Scientific literature demonstrates that an MTT assay can provide an estimation of the antiproliferative effect.

Dates from literature confirmed that *R. damascena* extract (ethanolic) had cytotoxic activity against the HeLa cell line and other studies indicated that *R. damascena* and its ingredients possess antitumor and anticancerogenic activity [44].

Regarding the cytotoxicity of nanoparticle preparations, of the three extracts, creams with lyophilized extract of *R. damascena* are best tolerated and cell viability is least affected by the use of this preparation. The most toxic effect on keratinocytes is apparently in the samples with extracts of *R. canina* L.

The cytotoxic activity of the synthesized *Rosa canina* Ag-nanoparticles against human colon adenocarcinoma cells HT29 was investigated by MTT assay and the IC50 value was found to be 7.89 µg/mL at 48 h incubation [45]. We have found in literature a study about the development of self-nanoemulsifying drug delivery systems for oil extracts of *Citrus aurantium* L. blossoms and *Rosa damascena* and evaluation of anticancer properties and the cytotoxicity effect of the formulations by the MTT assay indicated that the extract-loaded SNEDDSs represent enhanced anticancer activities, compared to the pure forms. All of the results demonstrated higher effectiveness of SNEDDS [33].

Using Franz diffusion test we showed the cumulative amount (%) of diffused quercetin in the different formulations as a function of time (minutes). The highest diffuse amount of quercetin was obtained in the formulations with SNEDDS in preparations containing *Rosa damascena* extract. Some researchers found that SNEDDS was able to increase the amount of α-mangostin which is transported through the stratum corneum over a certain time interval, which can be known by diffusion test [46]. The SNEDDS formula could improve absorption and bioavailability because nano-sized droplets can increase the release of insoluble drugs [47]. SNEDDS had relatively high lipid solubility to increase drug partitioning of the lipid membrane [48]. Nano preparations provided many benefits, such as increasing the absorption of herbal medicines and solubility compared to particles with larger sizes [49].

Plants with high amounts of antioxidants may be a promising therapy for preventing and curing UV-induced oxidative skin damage [50].

Superoxide dismutase (SOD), a free radical scavenging enzyme, is one of the first lines of cellular defense against oxidative injury [51,52]. Oxidative stress produces excessive reactive oxygen species (ROS), primarily due to imbalances in oxidative to reduction [53,54]. It has been shown that *Damask Rose* can reduce oxidative toxicity, and has a key role in disarming ROS [55]. In the present study, in vitro SOD activity is expressed as a percentage of SOD activity in untreated keratinocytes. *Rosa Damascena* extracts and SNEDDS formulation showed a significant increase in free radical scavenging activity and the highest SOD enzymatic activity compared to the controls. Most studies reported in the literature have focused on seed oils of other rose species, essential oils or fruit extracts, and less on the detailed study of petal extracts, especially rosehips.

Various herbal products having anti-psoriasis potential were tested in clinical studies and topical application of creams containing 10% *Mahonia aquifolium* extract [56], *Aloe vera* extract [57], ointment containing oleoresin from *Copeifera langsdorffii* (5%) [58], and cream with *Persea americana* oil [59] showed significantly greater improvements in psoriatic treatment compared with calcipotriol and fluticasone propionate mixture, hydrocortisone, triamcinolone acetonide, and calcipotriol ointment, respectively. In turn, studies with *Baphicacanthus cusia* ointment [60], *Curcuma longa* microemulgel [61], *Hypericum perforatum* ointment [62], and *Indigo naturalis* extract in oil [63], were found to be significantly more effective than the vehicle control group.

During our study, to verify the effectiveness of the treatment and to monitor the patients’ symptoms, we used for evaluation the PASI and DLQI score. After 6 weeks of use, all three indices (erythema, induration, scaling) of the assessment tool (PASI score) were reduced. The PASI, along with physician global assessment and quality of life measures, provide a complement of measures for studies of moderate to severe psoriasis that offer objectivity, are understandable to clinicians, and yield a comprehensive view of the impact of disease [64,65]. At the end of the 6 weeks of use of the cream, an improvement in the quality of life of the patients could be observed following the DLQI score. The DLQI is one of the most widely used QOL instrument in psoriasis-related clinical trials and has been used by various regulatory authorities and national dermatological associations to develop guidelines for the use of biologics [66]. Psoriasis can have a major impact on the quality of life (QOL) of patients by affecting their daily activities, social functioning and psychological well-being [67]. For patients who used the placebo, a slight reduction in scaling was observed at the first assessment. Erythema and induration were less reduced in contrast to patients who used the cream with extracts.

There are also literature data showing no significant difference between herbal products and drug/placebo treatment. Therefore, researchers are searching for new herbal products, which have the potential to be an alternative for synthetic drugs in psoriasis therapy, because long-term control of psoriasis using corticosteroids may cause skin atrophy, dermatitis, viral infection or other side effects [15].

Clinical trials reported in literature were carried out groups consisting of 10–20 patients, 30–50 patients, 60–80 patients, or more than 100 patients. The participants in these randomized, double-blind, placebo-controlled studies used either the topical cream with active substances or control (placebo) [68]. Other clinical trials included a nonrepresentative number of participants and therefore the conclusions of these studies cannot be generalized to the entire population affected by psoriasis. In clinical trials, the study protocols require many patient visits, including the initial one. At the first visit, a Psoriasis Area Severity Index (PASI) evaluation is performed by the doctor and each patient must complete a questionnaire that evaluates the quality life of the patient, for example, DLQI. Patients return at a predetermined time intervals after the beginning of treatment to be evaluated and at each visit patients complete the quality life index. The doctor evaluates the PASI score at each visit to verify the efficacy of the treatment. Compliance can be monitored via telephone calls and assistance is provided if it is needed.

Several factors might contribute to the explanation of discrepancies, for example a lack of standardization and quality control of the herbal extracts used in clinical trials, the use of different dosages; inadequate randomization in most studies and an improper selection of patients, with the numbers of them in most trials are insufficient for the attainment of statistical significance; wide variations in the duration of treatments using herbal medicines, and lacking or insufficient results of toxicological studies.

There are novel drug delivery systems such as liposome, niosome, ethosome, microemulsion, nanoemulsion, solid lipid nanoparticles, and nanostructured lipid carrier systems that can be used in extremely dehydrated and thickened psoriatic skin [69]. Liposomes offer an enhanced penetration through skin with less toxic effects compared to conventional drugs, niosomes slow down drug release and reduce systemic toxicity; moreover, ethosomes are able to reach the deep skin layers and/or the systemic circulation, microemulsions provide long-term stability and high solubilization capacity for hydrophilic and lipophilic drugs, prolonge action on the skin, and protect. Nanoemulsions protect the drug from instability, other systems ensure the compatibility of the drugs (NLCs) [69]. Topical drug delivery systems with plant extracts formulations enhance the therapeutic effects of the extracts and facilitate their penetration through the skin.

## 4. Materials and Methods

### 4.1. Preparation and Characterization of Dry Extract

Flower petals were collected from rose planters from Bihor county, Romania, in May 2020 and were dried at 70 °C for 2 h. The alcoholic extract solutions were prepared by maceration using 10 g of petals in 100 mL ethanol 70%, at room temperature for 7 days. The residue was removed by decantation. Then, the sample was kept at low temperature until the product was completely frozen and we obtained an lyophilized extract through a process of controlled freeze drying. In the stage of preformulation of the topical products, lyophilized extracts of roses were dissolved in Transcutol HP [70].

### 4.2. Formulation and Investigation of Self-Nanoemulsifying Drug Delivery System

Different self-emulsifying combinations have been formulated by the water and oil dilution method with a tenside (Cremophor RH 40) and a co-tenside (Transcutol HP). The compositions are listed in Table 6. The Transcutol HP and the Cremophor RH 40 were mixed at 37 °C by Schott Tritronic dispenser (SI Analytical, Mainz, Germany). The applied concentration of rose extract was dissolved in the system at room temperature by permanent agitation [71]. 

To evaluate any signs of phase separation, the mixtures were equilibrated for 24 h. A rotating paddle apparatus (Erweka GmbH, Heusenstamm, Germany) was used to measure the efficiency of self-emulsification. One gram of each mixture was added to 200 mL of distilled water with gentle agitation condition provided by a rotating paddle at 100 rpm and at a temperature of 37 °C. The process of self-emulsification was visually monitored for the rate of emulsification and for the appearance of the produced emulsions. The visual properties monitored against the growing of the surfactant component in Ternary triangular diagrams. Plotting points of the different compositions were selected according to cartesian coordinate calculation (Figure 1).

### 4.3. Formulation of Ointments Containing Lyophilized Rose Extracts and Rose-SNEDDS

To prepare a topical dosage form the mixture of cetostearyl alcohol, stearic acid and IPM were heated to 60 °C and homogenized. This formed the oily phase of the oil-in-water ointments. The mixture of propylene glycol, emulsifier agent (Sucrose stearate SP 70), and purified water was also heated to 60 °C, which formed the aqueous phase. The aqueous and oily phases were homogenized and cooled down to 25 °C. The final step was the addition of the active ingredient, the different rose extracts, in suspended (lyophilized) form or in a self-emulsifying drug delivery system. The composition of the ointments with rose extracts can be seen in Table 7.

### 4.4. Texture Analysis

Analysis of the textural properties was conducted in order to estimate the mechanical characteristics of the compositions. A compression test was performed and the resistance of formulations was measured by a CT3 Texture Analyzer (Brookfield, Middleboro, MA, USA). The force exerted on the probe was recorded using Texture Pro CT Software (Brookfield Engineering Laboratories, Middleboro, MA, USA). The texture analyzer was equipped with a TA5 cylinder-type probe (12.7 mm diameter and 35 mm length) during the test. The trigger load (5.0 g), target (10.0 mm), and speed (0.60 mm/s) of the device were fixed. Before measurement, the probe was lowered to the surface of the sample at a speed of 5 mm/s. After reaching the surface, the probe penetrated to a depth of 15.0 mm with a speed of 0.60 mm/s, and the force exerted on the probe was obtained. Both loading and unloading phases of the penetration curve were measured. The compression studies were performed at room temperature (24.5 ± 0.5 °C). All measurements were performed in triplicate. The average values and the standard deviation were calculated [72].

### 4.5. In Vitro Release Studies

Six Franz cells (Hanson Microette TM Topical and Transdermal Diffusion Cell System) were used for in vitro membrane permeation studies. The receptor cell was filled with 10.0 mL of PBS as the receptor phase and was stirred with a magnetic rotor, which was set to 300 rpm. For diffusion experiments, cellulose-acetate synthetic membranes (pore size 0.45 µm) were used, impregnated with IPM. Half a gram from each formulation was placed on the membrane. The effective diffusion surface area of each membrane was 1.767 cm^2^. The receptor solution was thermostated to 32.0 °C. The diffusion study was performed for 90 min. Samples of 1.0 mL were collected at predetermined time points of 15, 30, 60 and 90 min and replaced with the same quantity of the fresh receiving phase. The absorbance of the samples was measured at 370 nm. As a blank sample, PBS was used. A calibration curve of quercetin was determined before the spectroscopic measurements. A linear connection was found between the concentration of quercetin and the measured absorbance [73].

The quercetin release rate (k) was estimated from the slope of the amount of drug released per unit area (µg/cm^2^) versus the square root of time (min^½^). The diffusion coefficient (*D*) of the drug was determined from the drug concentration at a given t time (*Q*, µg/cm^2^), the initial concentration (C′0), and the diffusion time (*t*) (Equation (1)):(1)$$D=\frac{Q^{2}\times \pi }{{\left(2\left[C_{0}^{\prime }\right]\right)}^{2}\times t}$$

### 4.6. Superoxide Dismutase (SOD) Assay

For the antioxidant experiment, UV-B (Oriel^®^ Sol-UV-4 UV Solar Simulator, Bozeman, MT, USA) radiation was used to induce oxidative stress and free radical formation after the treatment. The antioxidant activity of formulations was determined on HaCaT cells after UV-B exposure. Samples (1.0 mL) were collected using the Franz diffusion device, as in the case of the MTT assay (1.0 mL). As positive control, HaCaT cells were treated with Trolox, which was dissolved in PBS immediately before use (15.0 µM). The cells were seeded on 12-well plates at a density of 10^5^ cells/well and grown in a CO_2_ incubator at 37 °C for four days. In the pretreatment group, culture medium was removed, 10 µL of the test solution was added, and the cells were incubated for a further 30 min with the samples. After 15 min of UV-B irradiation, samples were removed and cells were washed twice with PBS and incubated for 24 h. Cells were harvested using a rubber rod and centrifuged at 1000× *g* for 10 min at 4 °C. The cell pellet was then homogenized in 20 mM HEPES buffer (1 mM EGTA, 210 mM mannitol, and 70 mM sucrose/g tissue) and centrifuged at 10,000× *g* for 15 min at 4 °C and pH 7.2. The SOD activity of the supernatant was analyzed using assay kits from Cayman according to the manufacturer’s instructions (Cat. 706002, Cayman, Ann Arbor, MI, USA). All experiments were performed in triplicate [74].

### 4.7. Cell Viability Study (MTT Assay)

To evaluate cytotoxicity of the formulations, MTT assay was performed. The experiments were carried out on an HaCaT cell line. The cells had been maintained by weekly passages in Dulbecco’s DMEM culture media. For the assay, the cells were seeded on a 96-well plate, in the density of 10,000 cells/well. When they fully grew over the well’s membrane, the experiment was performed. First culture media was removed, then the test solutions were applied and the cells were incubated with them for 30 min. After 30 min, the test solutions were removed and MTT paint solution was added at 5 mg/mL concentration to the cells (tetrazolium-bromide). Then the cells were incubated with the MTT paint for 3 h. The viable cells will transform the water-soluble tetrazolium-bromide into formazan precipitate. When the incubation was completed, formazan precipitate was dissolved with the isopropanol: hydrochloride acid = 25:1 ratio. Then, the absorbance of these solutions was measured by spectrophotometer (Fluostar Optima, Debrecen, Hungary) and it’s directly proportional to the number of viable cells. The experiment was carried out with samples taken from the Franz diffusion chamber apparatus at predetermined times [75,76].

### 4.8. Clinical Study

To evaluate the efficacy of our formulations, 20 patients were recruited, of which 10 received a topical treatment that contained *Rosa damascena* extract in a self-emulsifying drug delivery system with a high content of active ingredients, and 10 patients received a placebo cream that was applied twice a day. During the study, in the first phase patients used a small amount, applied in a thin layer to be able to observe any side effects (e.g., hypersensitivity/potential allergies to cream compounds).

For each patient studied, psoriasis lesions with a similar size and the same clinical severity were selected. The clinical assessment of the severity of the psoriasis lesion was performed at the initial time, during, and at the end of the treatment using the scores of erythema, induration, and desquamation, on a scale from 0–4. Elements related to the patient’s quality of life were also assessed using DLQI score.

Creams were applied topically twice a day for 6 weeks, once in the morning and once in the evening, on psoriatic lesions; patients were evaluated at the beginning, after two weeks, and at the end of the study. For a better evaluation were taken photos were takenduring the monitored period. The study was carried out with the consent of the Ethics Commission CEMF/03 from 31 October 2021.

## 5. Conclusions

Considering the results of our studies regarding the extract of *R. damascena* Mill. and taking into account phytochemical screening and biological activity, we assessed the effect of topical formulation with SNEDDS of *R. damascena* Mill. on psoriatic patients. Because there are no studies in literature that used *Rosa damascena* Mill. extract on this skin affection, our development of topical drug delivery system facilitates plant extract dissolution and penetration through the skin and enhances the therapeutic effects of herbal products in psoriasis treatment. Because the patients’ feeling is the main focus, and our study showed that topical products containing *Rosa damascena* Mill. extract could be an complementary treatment to improve the quality life of the patient and it has fewer side effects than the conventional treatment used on long-term, more scientific evidence, and documentation is desired for the promotion of herbal treatment of psoriasis or other diseases, which must be substantiated by reliable clinical trials with standardized materials and formulations. Moreover, the growth of interest in natural medicine may force pharmaceutical companies to invest in extensive preclinical and well-controlled randomized clinical trials to prove the safety and efficacy of herbal medicines.

## Figures and Tables

**Figure 1 molecules-27-05499-f001:**
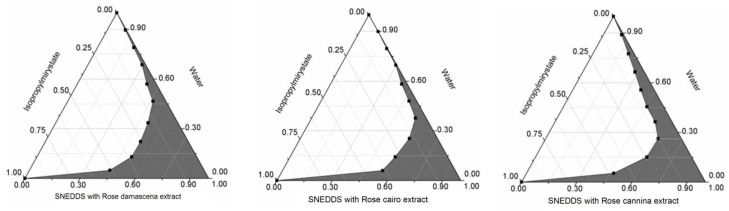
Pseudoternary phase diagrams of the different compositions. The dark area shows the nanoemulsion zone of the preparations.

**Figure 2 molecules-27-05499-f002:**
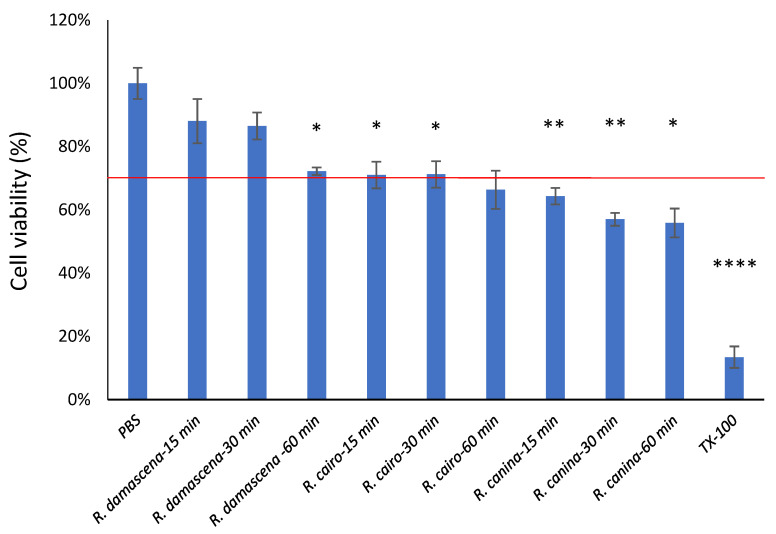
Cytotoxicity of different Rose SNEDDS samples on HaCaT cell line. Results of MTT assay of SNEDDS samples. *Rosa damascena* and *Rosa cairo* SNEDDS does not affect the cell viability because values are above 70%, however *Rosa canina* has a cytotoxic effect as cell viability is decreased below 70%. The results present the mean of 6 wells ± SD. Statistical analysis happened with one way ANOVA and *t*-test. Significant differences are marked in the figures with asterisks. * indicates significant differences at *p* = 0.0156; 0.0478; 0.0486; 0.0137. ** means significant differences at *p* = 0.0090; 0.023. **** means significant difference at *p* < 0.0001. The red line marks 70% of cell viability value, which is the recommendation of ISO 10993-5.

**Figure 3 molecules-27-05499-f003:**
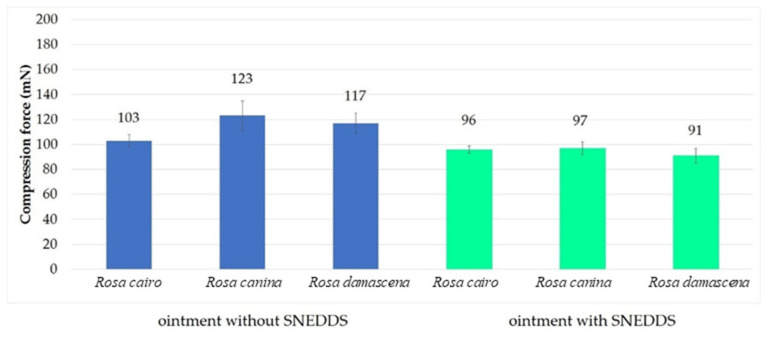
Results of the texture analysis of the formulated ointments with or without SNEDDS. Those ointment formulations, which contain the active ingredient in SNEDDS, demonstrated lower resistance values compared to those ones where rose was incorporated in suspended form.

**Figure 4 molecules-27-05499-f004:**
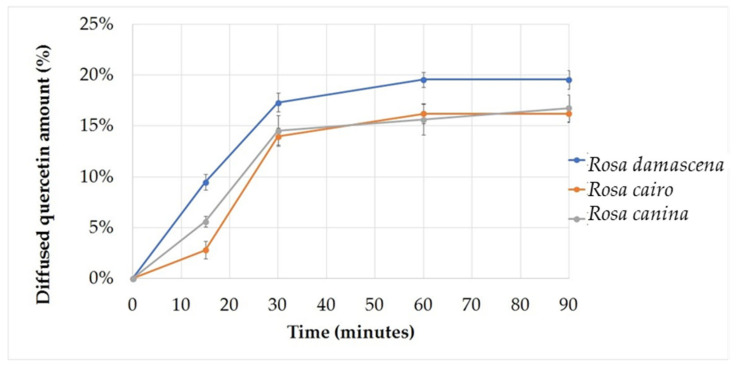
Diffusion profile of formulations without SNEDDS. The maximum amount of diffused quercetin reached 19.54% after 60 min, remaining at the same value after 90 min.

**Figure 5 molecules-27-05499-f005:**
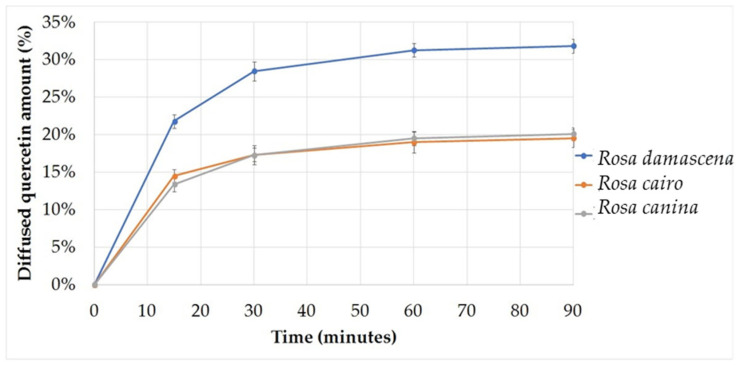
Diffusion profile of formulations with SNEDDS. The highest diffused amount of quercetin was obtained in the formulations with SNEDDS. The maximum diffuse amount of quercetin was 31.83% after 90 min.

**Figure 6 molecules-27-05499-f006:**
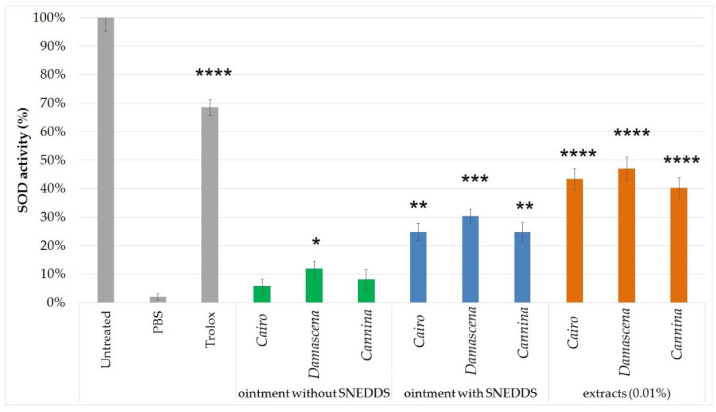
SOD activity of samples (green: without SNEDDS, blue: with SNEDDS, orange: extracts). SOD enzyme activity is expressed as the percentage of the enzyme activity of the cells without UV radiation (untreated). Data are expressed as mean ± SD and *n* = 6. Statistical significance is indicated *, **, ***, **** at *p* < 0.005, *p* < 0.01, *p* < 0.001, and *p* < 0.0001. Treated groups were compared to PBS with one-way ANOVA followed by Dunnett’s multiple comparison test.

**Figure 7 molecules-27-05499-f007:**
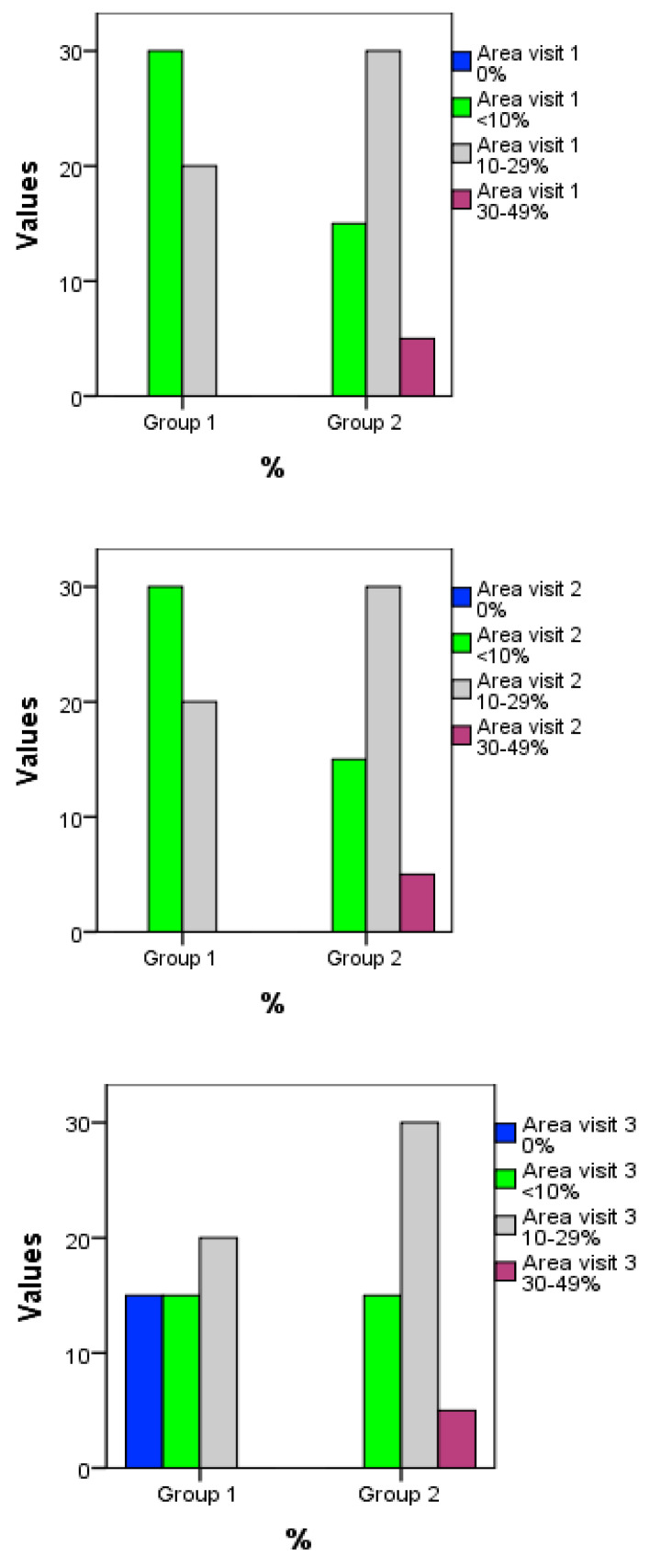
Graphic presentation of the percentage of the affected area in the 3 visits, at the two research groups. The treated group is represented by Group 1 and the placebo group is represented by Group 2.

**Figure 8 molecules-27-05499-f008:**
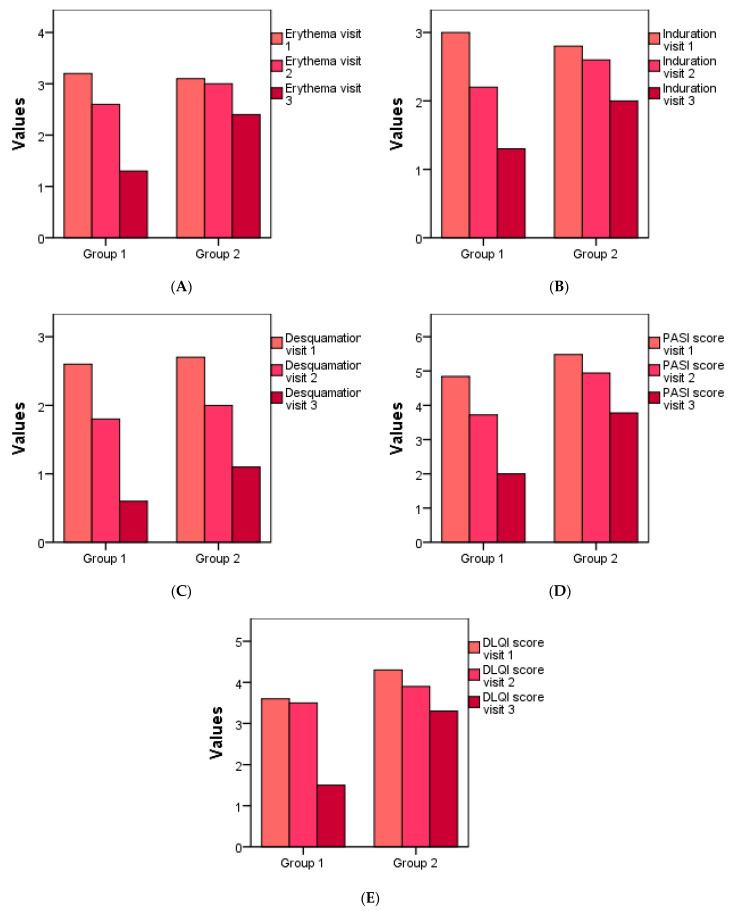
Nominal graphical presentation of the research parameters in the 3 visits, erythema (redness) (**A**), induration (thickness) (**B**), desquamation (scaling) (**C**), PASI Score (**D**), DLQI score (**E**), in the 2 research groups. The treated group is represented by Group 1 and the placebo group is represented by Group 2.

**Table 1 molecules-27-05499-t001:** Antioxidant capacity of alcoholic extracts.

Extract (Alcoholic 10%)	DPPH (%)	FRAP (μM TE/100 g DW)	CUPRAC (μM TE/100 mL)
*Rosa canina*	60.03 ± 0.01	844.24 ± 0.03	2077.67 ± 0.04
*Rosa cairo*	61.97 ± 0.03	850.12 ± 0.02	2055 ± 0.05

In this table antioxidant activity (means ± SD) of rose extracts is presented with different methods. Abbreviations: DW—Dry Weight, TE—Trolox Equivalent.

**Table 2 molecules-27-05499-t002:** Droplet size and zeta potential values of the formulated SNEDDS.

Composition	Droplet Size (nm)	Zeta Potential (mV)
SNEDDS–*Rose damascena*	91.75 ± 4.12	−33.1 ± 0.21
SNEDDS–*Rose cairo*	127.12 ± 3.22	−31.5 ± 0.18
SNEDDS–*Rose canina*	131.35 ± 5.21	−31.8 ± 0.24

**Table 3 molecules-27-05499-t003:** Quercetin release rate and the diffusion coefficient values related to the compositions.

Composition	Release Rate k 10^2^ (µg/cm^2×^ √min) ± S.D.	Diffusion Coefficient (after 90 min) D 10^5^ (cm^2^/min) ± S.D.
*R. damascena*	6.139 ± 0.05	0.146 ± 0.003
*R. cairo*	5.597 ± 0.09	0.156 ± 0.005
*R. canina*	5.894 ± 0.10	0.212 ± 0.011
*R. damascena* in SNEDDS	10.199 ± 1.32 **	0.213 ± 0.008 **
*R. cairo* in SNEDDS	6.436 ± 0.45 *	0.225 ± 0.006
*R. canina* in SNEDDS	6.622 ± 0.32 *	0.564 ± 0.012 **

Each data point represents the mean ± S.D., *n* = 5, and *p* < 0.05. Ordinary one-way ANOVA and Tukey’s multiple comparison test were performed to compare formulations with or without SNEDDS. ** Correlation is significant at the 0.01 level (2-tailed). Significant differences are marked with * in the table.

**Table 4 molecules-27-05499-t004:** Comparative statistics by the paired samples test method of the research parameters.

Paired Variables	Group 1				Group 2			
	Mean	SD	t	*p*	Mean	SD	t	*p*
Area_1—Area_3	0.30000	0.48305	1.964	0.081	0.30000	0.48305	1.964	0.081
Area_2—Area_3	0.30000	0.48305	1.964	0.081	0.30000	0.48305	1.964	0.081
Erythema_1—Erythema_2	0.60000	0.69921	2.714	0.024	0.10000	0.31623	1.000	0.343
Erythema_1—Erythema_3	1.90000	0.56765	10.585	0.001	0.70000	0.48305	4.583	0.001
Erythema_2—Erythema_3	1.30000	0.48305	8.510	0.001	0.60000	0.51640	3.674	0.005
Induration_1—Induration_2	0.80000	0.42164	6.000	0.001	0.20000	0.42164	1.500	0.168
Induration_1—Induration_3	1.70000	0.82327	6.530	0.001	0.80000	0.63246	4.000	0.003
Induration_2—Induration_3	0.90000	0.56765	5.014	0.001	0.60000	0.51640	3.674	0.005
Desquamation_1—Desquamation_2	0.80000	0.63246	4.000	0.003	0.70000	0.48305	4.583	0.001
Desquamation_1—Desquamation_3	2.00000	0.66667	9.487	0.001	1.60000	0.51640	9.798	0.001
Desquamation_2—Desquamation_3	1.20000	0.63246	6.000	0.001	0.90000	0.56765	5.014	0.001
PASI_Score_1—PASI_Score_2	1.12000	0.66800	5.302	0.001	0.54000	0.29889	5.713	0.001
PASI_Score_1—PASI_Score_3	2.84000	1.11076	8.085	0.001	1.70000	0.71336	7.536	0.001
PASI_Score_2—PASI_Score_3	1.72000	0.68767	7.909	0.001	1.16000	0.57966	6.328	0.001
DLQI_Score_1—DLQI_Score_2	0.10000	0.31623	1.000	0.343	0.40000	0.51640	2.449	0.037
DLQI_Score_1—DLQI_Score_3	2.10000	0.56765	11.699	0.001	1.00000	0.47140	6.708	0.001
DLQI_Score_2—DLQI_Score_3	2.00000	0.66667	9.487	0.001	0.60000	0.51640	3.674	0.005

The research parameters are presented from visit 1 to visit 3.

**Table 5 molecules-27-05499-t005:** Pearson correlation of the difference between research parameters.

Correlations							
Pearson Correlation		Area	Erythema	Induration	Desquamation	PASI Score	DLQI Score
Area	r	1	0.556 *	0.549 *	0.373	0.052	0.445 *
	*p*		0.011	0.012	0.105	0.828	0.049
Erythema	r	0.556 *	1	0.579 **	0.448 *	0.473 *	0.493 *
	*p*	0.011		0.007	0.048	0.035	0.027
Induration	r	0.549 *	0.579 **	1	0.704 **	0.564 **	0.346
	*p*	0.012	0.007		0.001	0.010	0.135
Desquamation	r	0.373	0.448 *	0.704 **	1	0.529 *	0.248
	*p*	0.105	0.048	0.001		0.017	0.292
PASI Score	r	0.052	0.473 *	0.564 **	0.529 *	1	0.502 *
	*p*	0.828	0.035	0.010	0.017		0.024
DLQI Score	r	0.445 *	0.493 *	0.346	0.248	0.502 *	1
	*p*	0.049	0.027	0.135	0.292	0.024	
	N	20					

* Correlation is significant at the 0.05 level (2-tailed). ** Correlation is significant at the 0.01 level (2-tailed).

**Table 6 molecules-27-05499-t006:** Compositions of the formulated SNEDDS containing different rose extracts.

SNEDDS	Rose Extract (*Rosa cairo/Rosa canina/Rosa damascena)*	IPM	Cremophor RH 40	Transcutol HP
Rose-SNEDDS	5 g	15 g	40 g	40 g

**Table 7 molecules-27-05499-t007:** Compositions of the formulated ointments containing different rose extracts.

	A	B	C	D	E	F
SP70	+	+	+	+	+	+
Cetostearyl Alcohol	+	+	+	+	+	+
Stearic acid	+	+	+	+	+	+
*R. cairo*	+	-	-	-	-	-
*R. canina*	-	+	-	-	-	-
*R. damascena*	-	-	+	-	-	-
*R. cairo* SNEDDS	-	-	-	+	-	-
*R. canina* SNEDDS	-	-	-	-	+	-
*R. damascena* SNEDDS	-	-	-	-	-	+
Isopropyl Myristate	+	+	+	+	+	+
Propylene Glycol	+	+	+	+	+	+
Distilled Water	+	+	+	+	+	+

+ sign means the component was incorporated into the formulation, - sign means it was not added to the given formulation.

## Data Availability

Data are available from the corresponding author with the permission of the head of the apartment. The data that support the findings of this study are available from the corresponding author (bacskay.ildiko@pharm.unideb.hu), upon reasonable request.

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
