# Peer review of "In Vitro and Human Pilot Studies of Different Topical Formulations Containing Rosa Species for the Treatment of Psoriasis"

_molecules, 2022, doi:10.3390/molecules27175499_

Round 1

Reviewer 1 Report

The paper describes development of various topical  formulations of Rosa species and a pilot study with the chosen formulation in the treatment of psoriasis.

The treatment improved the quality of life of the psoriatic patients, which is an important result, even regarding the rather low number of patients. The advantage of using SNEEDS in the topical formulaton is also demonstrated.

The paper seems to have three questions to address: i) which Rosa species to use, ii) which of the topical formulations have the best characteristics iii) the pilot clinical study.

The presentation of the results and the discussion needs some improvement.

As Rosa damascena extract was chosen because it showed the largest nanoemulsion existing zone, the detailed comparison of the three Rosa species is less important, especially as they show similar antioxidant capacity (although it is not known if this is the needed effect in psoriasis, since the inhibition of keratinocyte proliferation could also be an advantage). It is not discussed how the difference in self-nano-emulsifying capability of the three alcoholic extracts can be explained.

Figures 2-4 are recommended to be omitteed, since these results are clearly described in the text.

Legend of Fig. 5: significance level should be given in the usual way instead of indicating the calculated numbers. Rephrase the sentence: Rosa damascena and Rosa cairo SNEEDS are safe. The title of the Figure should be given in the legend.

Section 2.7, last sentence: Was the difference significant in SOD activity when the effect of the three Rosa SNEEDS were compared?

Table 4.: there is no comparison shown between the treated and placebo groups, and this way is not informative enough, it should be omitted.

Fig. 9 and 10: in the legends please indicate which is the treated and the placebo group.  

Table 5: Please give number in the last column where 0,000 is shown and please use decimal points.

Figure 11 is not needed. The same results are summarized in Table 6.

Discussion: the parts discussing the advantages of vegetables and fruits should be carefully re-written (especially the statements about their less side-effects, better compliance, etc). Discussing the composition of the Rosa extracts and their antioxidant properties is too detailed, should be more focused. (The comparison with previous results is difficult because they are not given in the same way – page 18, lines 377-378).

The paragraph of the MTT assay is not needed. (Or add that it can also show an antiproloferative effect).

The paragraph on the characteristics of clinical trials with natural products needs improvement.

Section 4.1.: Table 1 does not show the composition.

Section 4.8. Please clarify which topical formulation was used in the clinical study.

Conclusion: the first sentence shpold be omitted, because it is not correct this way.

I suggest the publication of the manuscript after the suggested improvements.

Author Response

Reply to the Reviewer 1

Dear Reviewer 1,

We would like to thank you for taking the time and effort to review the manuscript. We sincerely appreciate all your valuable comments and suggestions, which helped us in improving the quality of the manuscript. We would like to take this opportunity to thank you for the effort and expertise that you contributed towards reviewing the article. Our revision in the manuscript was highlighted by red colour. Some observations were similar to the Reviewer 2 and were highlighted by green colour.

“As Rosa damascena extract was chosen because it showed the largest nanoemulsion existing zone, the detailed comparison of the three Rosa species is less important, especially as they show similar antioxidant capacity (although it is not known if this is the needed effect in psoriasis, since the inhibition of keratinocyte proliferation could also be an advantage). It is not discussed how the difference in self-nano-emulsifying capability of the three alcoholic extracts can be explained.”

After establishing the components of SNEDDSs, optimization studies are conducted to obtain the best amounts of oily phase, surfactants and cosolvents that generate spontaneous nano-emulsion. Ternary phase diagrams are used for the identification of the emulsification area using the selected components. This area can be identified visually or by measuring the particle size of the nano-emulsion that results after the aqueous dispersion. It was shown that the SNEDDSs composition from the emulsification area generate spontaneous nano-emulsions that has globule sizes less than 200 nm. After aqueous dispersion we can observe a region with homogenous and clear systems that is represented by the region of nano-emulsification. According to our diagrams the most favorable formulation is the one which contained R. damascena with the largest nanoemulsion existing zone and the smallest droplet size (91.75±4.12 nm).

“Figures 2-4 are recommended to be omitteed, since these results are clearly described in the text.”

Thank you for your observation. Figures 2-4 were omitted.

“Legend of Fig. 5: significance level should be given in the usual way instead of indicating the calculated numbers. Rephrase the sentence: Rosa damascena and Rosa cairo SNEEDS are safe. The title of the Figure should be given in the legend.”

The figures were renumbered after removing those ones that were irrelevant to the article and the title of the Figure was given in the legend. The calculated numbers were removed, significance level is presented in the text. The sentence „Rosa damascena and Rosa cairo SNEEDS are safe” was rephrased.

Figure 2. Cytotoxicity of different Rose SNEDDS samples on HaCaT cell line. Results of MTT assay of SNEDDS samples. Rosa damascena and Rosa cairo SNEDDS does not affect the cell viability because values are above 70%, however Rosa canina has cytotoxic effect as cell viability is decreased below 70%.

“Section 2.7, last sentence: Was the difference significant in SOD activity when the effect of the three Rosa SNEEDS were compared?”

Pretreatment with the ointment containing the R. damascena in SNEDDS showed the highest SOD enzymatic activity (30%) as we can observe on the statistical significance (p< 0.001) compared to the other rose extracts (SOD activity <30%; p< 0.0001).

“Table 4: there is no comparison shown between the treated and placebo groups, and this way is not informative enough, it should be omitted.”

Thank you for your suggestion. Table 4 was omitted.

“Fig. 9 and 10: in the legends please indicate which is the treated and the placebo group.“

The placebo and the treated group were clarified. Group 1 is the treated group and Group 2 is the placebo one.

“Table 5: Please give number in the last column where 0,000 is shown and please use decimal points.”

Thank you for the suggestion, numbers are given now and decimal points are used.

“Figure 11 is not needed. The same results are summarized in Table 6.”

Thank you for our observation. Figure 11 was omitted. It is highlighted with green.

“Discussion: the parts discussing the advantages of vegetables and fruits should be carefully re-written (especially the statements about their less side-effects, better compliance, etc). Discussing the composition of the Rosa extracts and their antioxidant properties is too detailed, should be more focused. (The comparison with previous results is difficult because they are not given in the same way – page 18, lines 377-378).”

The statement about the advantages of vegetables was removed and was specified that researchers are looking for new herbal products, which have the potential to be an alternative for synthetic drugs in psoriasis therapy, because long-term control of psoriasis using corticosteroids may cause skin athropy, dermatitis, viral infection or other side-effects. The discussion about the composition and antioxidant properties of Rosa extracts was reviewed and irrelevant statements were removed.

 Qualitative, quantitative characterization also antioxidant activity for the alcoholic extract of Rosa damascena can be consulted on our previous research article as phytochemical screening and biological activity show that the extract is a wide range of natural source which can be used for the discovery of natural pharmaceuticals and due to the high content of bioactive compounds (TPC=321 ± 0.03 mg GAE/100 g DW and TFC=32.4 ± 0.01 mg QE/100 g DW) and high values for antioxidant activity: DPPH (94.11 ± 0.01%), CUPRAC (555.316 ± 0.06 µmol Trolox/mL), FRAP (354.43 ± 0.03 µmol Trolox/mL) and ABTS (76.351 ± 0.02 µmol Trolox/mL), it can be used as prophylactic agents or as an adjuvant treatment in the pathologic process of various disorders. (Highlighted with green)

“The paragraph of the MTT assay is not needed. (Or add that it can also show an antiproloferative effect).”

Thank you for the recommendation, it was added that Scientific literature demonstrate that MTT assay can provide an estimation of the antiproliferative effect.

“The paragraph on the characteristics of clinical trials with natural products needs improvement.”

            Thank you for the suggestion, the paragraph is rewritten: In clinical trials, the study protocols requires many patient visits, including the initial one. At the first visit, a Psoriasis Area Severity Index (PASI) evaluation is performed by the doctor and each patient must complete a questionnaire that appreciates the quality life of the patient, for example DLQI. Patients return at a predetermined time intervals after the beginning of treatment to be evaluated and at each visit patients complete the quality life index. The doctor evaluates the PASI score at each visit to verify the efficacy of the treatment. Compliance can be monitored via telephone calls and assistance is provided if it's needed.

“Section 4.1.: Table 1 does not show the composition.”

Thank you for the observation, the compositions of different self-emulsifying combinations are listed in Table 6.

“Section 4.8. Please clarify which topical formulation was used in the clinical study.”

Thank you for the comment, in the clinical study the topical formulation with Rosa damascena extract in self-emulsifying drug delivery system was used.

“Conclusion: the first sentence should be omitted, because it is not correct this way.”

Thank you for the suggestion, the first sentence was omitted from the conclusion.

Thank you for your very careful review of our paper, for the comments and suggestions!

Reviewer 2 Report

This paper presents the studies on major chemical constituents of three Rosa species and their antioxidant activities and preliminary clinical antipsoriasis treatment. Unfortunately, the research results have not been stated correctly in abstract (line 34-37) and conclusion. There are many mistakes in the manuscript, especially the langage needs to be polished, such as line 106...... Research parameters in tables and figures need to be explained clearly. Data needs to be in correct form, use period not comma as decimal point. In the tables, the SDs of clinical study data were too big, which needs to be intepreted carefully. The major phenolic compouds and their contents in Rosa damascena of previous study need to be given out as a comparison. Rosa cairo is a species not a variety. Plant's scientific names need to be in italic form and the first letter of their species words should always be in small letter form, as "cairo".

Author Response

Reply to the Reviewer 2.

Dear Reviewer 2,

We would like to thank you for taking the time and effort to review the manuscript. We sincerely appreciate all your valuable comments and suggestions, which helped us in improving the quality of the manuscript. We would like to take this opportunity to thank you for the effort and expertise that you contributed towards reviewing the article. Our revision in the manuscript was highlighted by green colour.

“This paper presents the studies on major chemical constituents of three Rosa species and their antioxidant activities and preliminary clinical antipsoriasis treatment. Unfortunately, the research results have not been stated correctly in abstract (line 34-37) and conclusion. There are many mistakes in the manuscript, especially the langage needs to be polished, such as line 106...... Research parameters in tables and figures need to be explained clearly. Data needs to be in correct form, use period not comma as decimal point. In the tables, the SDs of clinical study data were too big, which needs to be intepreted carefully. The major phenolic compouds and their contents in Rosa damascena of previous study need to be given out as a comparison. Rosa cairo is a species not a variety. Plant's scientific names need to be in italic form and the first letter of their species words should always be in small letter form, as "cairo".”

  • Thank you for the suggestion, the abstract was revised and part of results were stated differently. The aim of this study was to evaluate the phytochemical profile and antioxidant properties of the extracts from three Rosa species ( caninaR. damascenaR. cairo), to develop and investigate topical formulations with lyophilized forms of extracts for the treatment of psoriasis. After the evaluation of topical products, the formulation with Rosa damascena extract in self-emulsifying drug delivery system was tested on a human clinical study that involved 20 patients. At the end of the clinical study an improvement in the quality of life of the patients was observed and erythema, induration and scaling were reduced.

The present study indicates that our examined extracts exhibited great phenolic content, antioxidant capacity and safety profile of topical formulation and therefore can be used as a reliable source of natural antioxidants and may be used as a complementary treatment to improve the quality life of the patients with psoriasis or may be tested on another diseases.

  • The conclusion was revised and regarding the affirmations, were taking into account only our research results.
  • Thank you for the recommendation, the language of the manuscript was checked by a native English speaker colleague of ours, to make sure mistakes are avoided.
  • Thank you for your comment, line 106 was reformulated.
  • Thank you for the recommendation, research parameters in tables and figures are rewritten and explained more clearly. Tables were revised and was used period as decimal point. In the tables, from a statistical point of view, SDs represent the fluctuation between the minimum and maximum of the research variables, and when it exceeds the minimum, it means that the lot is not a compact one, but the distribution between the minimum and maximum value is large.
  •  
  • Thank you for your comment, the paraghraph was improved. Qualitative, quantitative characterization also antioxidant activity for the alcoholic extract of Rosa damascena can be consulted on our previous research article as phytochemical screening and biological activity show that the extract is a wide range of natural source which can be used for the discovery of natural pharmaceuticals and due to the high content of bioactive compounds (TPC=321 ± 0.03 mg GAE/100 g DW and TFC=32.4 ± 0.01 mg QE/100 g DW) and high values for antioxidant activity: DPPH (94.11 ± 0.01%), CUPRAC (555.316 ± 0.06 µmol Trolox/mL), FRAP (354.43 ± 0.03 µmol Trolox/mL) and ABTS (76.351 ± 0.02 µmol Trolox/mL), it can be used as prophylactic agents or as an adjuvant treatment in the pathologic process of various disorders.
  • Thank you for the observation, Rosa cairois a species not a variety. The statement was reviewed and modified.
  • Thank you for the suggestion, plant's scientific were written in italicform and the first letter of their species words were also written in small letter form.

Thank you for your very careful review of our paper, for the comments and suggestions!
